# Silicon-Nanowire-Type Polarization-Diversified CWDM Demultiplexer for Low Polarization Crosstalk

**DOI:** 10.3390/nano13162382

**Published:** 2023-08-21

**Authors:** Seok-Hwan Jeong, Heuk Park, Joon Ki Lee

**Affiliations:** 1Department of Electronic Materials Engineering, The University of Suwon, Hwaseong 18323, Republic of Korea; 2Electronics and Telecommunications Research Institute (ETRI), Daejeon 34129, Republic of Korea; parkh@etri.re.kr (H.P.); juneki@etri.re.kr (J.K.L.)

**Keywords:** silicon-nanowire waveguide, optical filter, optical demultiplexer, polarization control, O-band, coarse wavelength division multiplexing

## Abstract

Coarse wavelength division multiplexing (CWDM)-targeted novel silicon (Si)-nanowire-type polarization-diversified optical demultiplexers were numerically analyzed and experimentally verified. The optical demultiplexer comprised a hybrid mode conversion-type polarization splitter rotator (PSR) and a delayed Mach–Zehnder interferometric demultiplexer. Si-nanowire-based devices were fabricated using a commercially available Si photonics foundry process, exhibiting nearly identical spectral responses regardless of the polarization states of the input signals under the PSR. The experiment demonstrated a low insertion loss of 1.0 dB and a polarization-dependent loss of 1.0 dB, effectively suppressing spectral crosstalk from other channels by less than −15 dB. Furthermore, a TM-mode rejection-filter-integrated optical demultiplexer was designed and experimentally validated to mitigate unwanted TM-mode-related polarization crosstalk that arose from the PSR. It exhibited an improved polarization crosstalk rejection efficiency of −25 dB to −50 dB within the whole CWDM spectral range.

## 1. Introduction

Four-channel coarse wavelength division multiplexing (CWDM) in the O-band spectral range is widely used in datacenter applications owing to its cost-effectiveness [1,2,3,4]. In optical transceivers, optical demultiplexers are essential for filtering WDM optical signals and spatially separating them into output ports at the receiver end. These optical demultiplexers are typically fabricated on silicon-on-insulator (SOI) platforms, which provide technical advantages such as small footprint, cost-effectiveness, and monolithic integration, along with other functional devices such as optical modulators and detectors [2,3,5,6]. The operation of optical demultiplexers is commonly based on lightwave interferences in microring resonators [7,8,9], multistage delayed Mach–Zehnder interferometers (DMZI) [10,11,12,13,14,15,16,17], and arrayed waveguide gratings (AWGs) [18,19,20,21].

CWDM systems offer several advantages over dense WDM systems because of their relaxed laser oscillation wavelength accuracy and wider operating wavelength range. In addition, CWDM optical devices must operate within a spectral range of more than 60 nm because the channel grid in the wavelength domain is spaced by 20 nm [1].

Meanwhile, the polarization state of an optical signal is not maintained because it propagates through standard single-mode fibers (SMFs). Consequently, the optical demultiplexer on the receiver’s side must be capable of handling arbitrary polarization input signals [13,15,21,22]. Various approaches have been explored to handle arbitrarily polarized WDM signals at the receiver end. These approaches are classified into two categories: polarization-independent and polarization compensation schemes, or polarization diversity schemes for silicon nitride (SiN) [11,13,21] and silicon (Si) [7,8,9,10,12,14,15,16,17,18,19,20,22] materials.

The polarization-independent scheme has been widely used in SiN material systems [13,21] because Si-nanowire-based waveguides primarily exhibit large birefringence. Square-shaped cross-sectional dimensions have been used to depolarize the split/coupling response and equivalent refractive index of SiN waveguides [13]. Despite experimental demonstrations of the polarization-independent scheme on SiN materials, there is still a lack of spectral uniformity between the two orthogonal polarization input signals (TE and TM modes). The proposed approach requires an additional coupling scheme to optically connect to other Si-nanowire-based functional building blocks, such as Si-nanowire waveguides and optical detectors, because of the relatively high refractive index contrast [23].

In addition, polarization-compensated AWG has been utilized in the SiN material platform for CWDM applications [21]. This device can operate irrespective of the input polarization state because of the multiple polarization rotators based on laterally asymmetric waveguides in the middle of each arrayed waveguide of the AWG. However, further investigation is required to address concerns such as the large footprint, the relatively complex fabrication process of the polarization rotators, and the lack of monolithic integration with other Si-nanowire-based functional devices.

Si-nanowire-based optical demultiplexers have been actively investigated because of their small size and monolithic integrability. The polarization problem caused by the relatively large birefringence of Si-nanowire waveguides has been addressed using two different schemes: the polarization compensation scheme for the CWDM-targeted four-channel DMZI device [15] and the polarization diversity scheme for the combination of the DMZI interleaves and eight-channel AWGs [22]. In the DMZI scheme [15], several polarization rotators based on laterally asymmetric waveguide shapes were placed in the center of all single-stage DMZIs. These rotators interchange the polarization state whenever the signal propagates through each DMZI, enabling the device to depolarize the spectral response regardless of the polarization state of the input signal. This is achieved using a Si core thickness of 340 nm to depolarize the optical split and coupling response of the bent-shaped directional coupler. However, using a thicker Si core (than the standard value of 220 nm) may cause issues in lowering the propagation loss of the Si-nanowire waveguides. Furthermore, the polarization rotators require high accuracy in the lithography process (e.g., electron beam lithography), making them unsuitable for high-volume production. Reference [22] described a polarization-diversity-based 16-channel WDM filtering scheme that utilized a polarization splitter rotator (PSR) and two types of optical demultiplexers. However, because the DMZI-type optical filter was designed to operate within a wavelength range of 30 nm, applying this scheme to the CWDM-targeted optical demultiplexer posed a challenge.

Overall, as stated earlier, from the viewpoint of polarization-insensitive CWDM optical demultiplexers, previously reported SiN-based devices showed a lack of spectral uniformity [13], a large footprint of more than mm-order length and a complicated fabrication process [21], and difficulty in monolithic integration with Si-nanowire-based devices [13,21,23]. Meanwhile, Si-nanowire-based devices had concerns about higher propagation loss and higher fabrication accuracy caused by a thicker Si-core layer [15]. Consequently, few reports have been published on polarization-insensitive CWDM optical demultiplexers based on Si-nanowires for highly efficient optical receivers. To address the technical drawbacks mentioned above, we introduced a polarization-diversity-type CWDM optical demultiplexer based on Si-nanowires [24]. Although the fabricated device demonstrated polarization-diversified operation with a low excess loss of less than 1.0 dB and a low polarization-dependent loss of less than 1.0 dB, a certain level of TM-mode-related polarization crosstalk persisted. To minimize the polarization crosstalk mentioned earlier, we developed a new TM-mode rejection filter-integrated polarization-diversified CWDM optical demultiplexer based on Si-nanowire waveguides.

In this study, we proposed, theoretically analyzed, and experimentally demonstrated a novel Si-nanowire-based optical demultiplexer targeting CWDM in the O-band regime. Section 2 presents the demultiplexer structure, comprising a hybrid mode conversion-type PSR, TM-mode rejection filters, and two identical DMZI-type four-channel cascade-connected filters. Subsequently, numerical simulation results of the operational characteristics are described. Section 3 depicts the device fabrication, experimental characterization, and remaining technical issues to be addressed. We successfully suppressed the residual polarization crosstalk of the fabricated demultiplexer by more than −25 dB across the entire CWDM spectral range by integrating with the TM-mode rejection filters while maintaining the CWDM optical demultiplexer functionality.

## 2. Device Structure and Theoretical Considerations

Figure 1 illustrates the schematic of the polarization-diversified CWDM optical demultiplexer. This device includes the PSR, which provides polarization diversity; the TM_00_-mode rejection filter, which minimizes polarization crosstalk; and two identically designed DMZI-type optical demultiplexers for filtering CWDM optical signals. The PSR consists of a hybrid mode conversion region based on rib-shaped Si-nanowire waveguides and a mode split region based on a rectangular-shaped Si-nanowire asymmetric directional coupler (ADC) (Figure 1). Typically, linearly polarized arbitrary states, such as the combination of TE_00_-mode and TM_00_-mode, can be split into TE_00_-mode outputs and sent to spatially separated channels within the ADC [22,24].

As stated in Ref. [25], the hybrid mode conversion-type PSR utilizes shallow-etched rib and normal channel waveguides, which are easily fabricated using the currently available Si photonics foundry process [26]. The PSR employs rib-shaped waveguide structures with a thin slab width set at 100 nm to implement hybrid mode conversion between the TM_00_-mode and the TE_01_-mode in the rib waveguide region. Here, the optimized thin slab width of 100 nm was narrow enough to excite only the hybrid modes of TM_00_-mode and TE_01_-mode at the rib waveguide width of 550 nm. To maximize the mode conversion efficiency, the rib waveguide width along the propagation direction was optimized on the basis of numerical simulation results from the finite element method (FemSIM^TM^, Synopsys, Sunnyvale, CA, USA). More details of the numerical simulation can be found in our previous work [24]. Regardless of the input signal’s polarization state, only the TM_00_-mode was transferred to the TE_01_-higher-order mode (Figure 1). The two propagating modes, TE_00_-mode and TE_01_-mode (Figure 1), were spatially separated through the mode split region based on the ADC.

In this study, we designed, numerically simulated, and experimentally verified two types of ADC schemes. Figure 2 shows the calculated propagation characteristics for the two types of ADC-based mode split regions in the PSR: (a) the top view of the mode split area based on a long-length ADC (L-ADC), (b) the specification of the linear-taper-based L-ADC, (c) the simulated optical intensity profile along the L-ADC region (λ_in_ = 1.31 μm), (d) the top view of the mode split area based on a short-length ADC (S-ADC), and (e) the simulated optical intensity profile along the S-ADC region (λ_in_ = 1.31 μm). A packaged software based on a three-dimensional (3D) beam propagation method (BeamPROP^TM^, 2021.03, Synopsys) was used for implementing the simulation shown in Figure 2c,e. In the case of L-ADC, the value of W_Gap_ was chosen to facilitate fabricating in the currently available foundry process, while the values of W_A_, W_B1_, and W_B2_ were set to facilitate coupling the TE_01_-mode to the neighboring waveguide. Meanwhile, in the case of S-ADC, the values of W_C_ and W_D_ were adjusted to experience efficient directional coupling for the set value of W_Gap_.

Coupling the TE_00_-mode light to a neighboring waveguide is difficult because of the significant equivalent index mismatch between the two coupled waveguides in the ADC scheme. In both cases shown in Figure 2 (L-ADC and S-ADC), optimizing the relatively narrow waveguide width allows for index matching between TE_00_-mode and TE_01_-mode, enabling the coupled TE_01_-mode light converted from TM_00_-mode to be directed to the adjacent output channel as TE_00_-mode. Consequently, arbitrary polarization input signals can be processed as TE_00_-mode output in the PSR (Figure 2c,e).

The L-ADC shown in Figure 2b uses a linearly tapered scheme in the narrow waveguide. This approach allows various wavelengths to satisfy the equivalent index matching by gradually increasing the waveguide width from W_B1_ to W_B2_ along the L_L-ADC_ of 300 μm. Consequently, the PSR’s operating range in the wavelength domain could be broader. In simpler terms, the L-ADC scheme is more tolerant to coupled waveguide parameter variations caused by fabrication imperfections. Referring to Figure 2c, when using the parameters shown in Figure 2b, only the TE_01_-mode light tends to couple to the neighboring channel, while the TE_00_-mode light passes through the ADC without any mode coupling.

The S-ADC shown in Figure 2d does not use any tapered structure. The coupling length (L_S-ADC_) can be reduced to 12 μm by optimizing the coupled waveguide parameters, like W_C_, W_D_, and W_Gap_, for a specific wavelength value (i.e., λ = 1.31 μm) which is 25 times shorter than the L-ADC’s case. However, one of the potential drawbacks of the S-ADC scheme is its relatively smaller fabrication tolerance. In the PSR, the hybrid mode conversion efficiency mentioned above typically has a finite value, causing each converted TE_00_-mode signal to inherently include some residual TM_00_-mode components. This polarization crosstalk related to the TM_00_-mode is undesirable because it affects the receiver’s detection efficiency. To address this issue, polarization crosstalk components can be selectively removed using TM_00_-mode rejection filters, which rely on cascade-connected symmetric directional couplers (SDCs). Figure 3 shows (a) a schematic of the TM_00_-mode rejection filter based on the SDC and (b) a numerical simulation of the optical intensity profile along the sixth cascade-connected SDC region (λ_in_ = 1.31 μm). Unlike the ADC, the SDC scheme is normally unable to prevent the coupling of the TE_00_-mode to the neighboring waveguide. However, the coupling efficiency of the TE_00_-mode can be minimized when the equivalent index difference between the TE_00_-mode and the TM_00_-mode is maximized by adjusting the parameters of the coupled waveguides (i.e., W_E_, W_F_, W_Gap_), which makes the SDC work as the TM_00_-mode rejection filter by discarding the directional coupling components, as shown in Figure 3a.

Considering the above-mentioned design guideline, the SDC parameters were set to W_E_ = W_F_ = 350 nm and W_Gap_ = 400 nm, as shown in Figure 3a. Typically, the equivalent refractive index for the TM_00_-mode is smaller than that of the TE_00_-mode [25,27]. Consequently, the TM_00_-mode can be easily coupled to the neighboring waveguide, while the TE_00_-mode is unaffected by mode coupling because of the optimized coupling length (L_SDC_ = 20 μm). The propagation characteristics simulated by using a 3D beam propagation method (BeamPROP^TM^, Synopsys) are shown in Figure 3b. We devised a filtering scheme consisting of a sixth cascade connection of the SDCs (Figure 1) to maximize rejection efficiency for the TM_00_-mode and minimize the excess loss for the TE_00_-mode. Thus, TM_00_-mode crosstalk was reduced by 15 dB with a low excess loss of 0.6 dB for the main signal.

The PSR emits TE_00_-mode output signals, which are directed toward each DMZI-based optical demultiplexer (Figure 1). The direction of the signals depends on their initial polarization state before entering the PSR. The optical demultiplexer comprises three optical delays, with each path difference carefully optimized to achieve constructive interference around the CWDM channel grid. To operate within a CWDM spectral range of more than 60 nm, we used two identical 3 dB multimode interference (MMI) couplers in each DMZI rather than the SDC schemes with sinusoidal coupling behaviors [28]. The rectangular-shaped MMI region was designed based on the 2:2 general interference mechanism from the viewpoint of the compact size of the MMI region [29].

Except for the MMI region, the Si-nanowire waveguide width in the optical demultiplexer was set to 350 nm. It is crucial to properly adjust the path differences at each DMZI, such as 2ΔL, ΔL, and ΔL′ (=ΔL + 0.5π -radian phase shift), to meet the CWDM filtering grids for each output channel and achieve the specified channel spacing of 20 nm in the O-band regime (Figure 1). To determine the appropriate value of ΔL, factors such as the equivalent refractive index and the group index of the 350 nm wide Si-nanowire waveguide were considered. ΔL was ultimately set to 5 μm. Refer to Reference [24] for more details on the design guidelines.

The optical demultiplexers efficiently process the two orthogonal polarization states of the signal, offering low insertion loss and crosstalk. The PSR is crucial for the proper handling of arbitrarily polarized light, and the MMI coupler should maintain a balanced optical split ratio close to 50:50. The two identical optical demultiplexers are symmetrically shown in Figure 1, facilitating the optical coupling of their outputs to a single photodiode (PD) array where the two outputs are counter-propagated to the single PD.

## 3. Fabrication and Experimental Characterizations

The CWDM optical demultiplexers were fabricated on a 200 mm waferscale using commercially available 193 nm ArF-dry lithography process technology, with a 220 nm thick Si core layer and a 2 μm thick buried oxide layer [26]. The parameters used for the waveguides in the PSR and the DMZI-based optical demultiplexers were identical to those described in the numerical calculations. The fabrication condition was managed enough to match with the theoretically defined parameters. Of course, there could be some portion of deviation between the design and fabrication, which is not consistent across the whole SOI wafer, from wafer to wafer, and from fabrication run to run. It is quite difficult to accurately measure the width or height of the fabricated waveguide parameters by using, i.e., a scanning electron microscope, as long as the fabricated chip is not physically destroyed. It is noted that several kinds of pattern specifications (i.e., waveguide width and gap) were measured using another pilot wafer for monitoring. As for the 350 nm wide waveguide samples, the deviation of the fabricated waveguide width was calibrated to within ±5 nm.

A tunable laser with a tuning range set from 1270 to 1330 nm was used as the light source to measure the transmission spectra of the fabricated devices. An optical vector analyzer (OVA, LUNA 5013) was used to characterize the transmission spectra of the tested chips. This analyzer calibrates the amplitude and phase information of the signal simultaneously. The Jones matrix of the output signals was analyzed to obtain the transmission spectra via optical interactions from arbitrarily polarized input to various output modes (e.g., TE_00_-mode input to TE_00_-mode output, TM_00_-mode to TE_00_-mode, TE_00_-mode to TM_00_-mode, and TM_00_-mode to TM_00_-mode) through SMFs [30,31].

Figure 4 shows the top view of the fabricated polarization-diversified Si-nanowire optical demultiplexers without the TM_00_-mode rejection filter (a) and with the TM_00_-mode rejection filter (b). The sole distinction between the two device structures is the presence or absence of the TM_00_-mode rejection filter. The entire scheme shown in Figure 4b is identical to that shown in Figure 1. To account for the accuracy margin in the fabrication process of a foundry, we utilized the L-ADC design shown in Figure 2 in the mode split region for both cases depicted in Figure 4. Depending on the presence of external filters, the chip size measures 700 µm long and 210 µm wide in Figure 4a and 950 µm long and 210 µm wide in Figure 4b. Note that the chip size could be further reduced because the device design was not optimized for a small footprint. Dummy patterns were integrated into the waveguide layer to enhance the process stability during fabrication (Figure 4).

First, we characterized the transmission characteristics of the fabricated device (Figure 4a). The measured spectra for all output channels ranging from Ch-1 to Ch-8 are shown in Figure 5. The optical vector analyzer provided the following measured spectral responses in Figure 5a–d: (a) main signals (TE_00_-mode input to TE_00_-mode output for Ch-1–Ch-4, TM_00_-mode input to TE_00_-mode output for Ch-5–Ch-8), (b) cross-coupled crosstalk (TM_00_-mode input to TE_00_-mode output for Ch-1–Ch-4, TE_00_-mode input to TE_00_-mode output for Ch-5–Ch-8), (c) TE_00_-to-TM_00_-mode polarization crosstalk (TE_00_-mode input to TM_00_-mode output for Ch-1–Ch-8), and (d) TM_00_-to-TM_00_-mode polarization crosstalk (TM_00_-mode input to TM_00_-mode output for Ch-1–Ch-8).

Figure 5a shows the experimental verification of the polarization-insensitive demultiplexing response. The output channels ranging from Ch-1 to Ch-4 exhibited sinc-function-shaped filter spectra, spaced by 20 nm, indicating that the TE_00_-mode input signal was spatially separated from the upper output of the PSR without any mode conversion and subsequently launched into the upper-side optical demultiplexer. On the other hand, the TM_00_-mode input signals were detected at output channels ranging from Ch-5 to Ch-8. Despite the severe birefringence of the 350 nm wide Si-nanowire waveguide, the measured spectra were almost identical to those from Ch-1 to Ch-4, confirming that only the TM_00_-mode experienced hybrid mode conversion to TE_01_-mode, with its mode being split as the TE_00_-mode in the L-ADC. As a result, arbitrarily polarized signals were properly demultiplexed within the CWDM spectral range, demonstrating a low excess loss of less than 1.0 dB and a low polarization-dependent loss (PDL) of less than 1.0 dB. In this case, the excess loss refers to the additional loss generated in comparison to the propagation loss of an S-bend-shaped waveguide with the same length. This excess loss accounts for losses resulting from a hybrid mode conversion, mode split, and MMI coupling behaviors. The spectral crosstalk from neighboring channels was measured to be approximately −20 dB for the peak filtering grid of each four-output channel.

The measured spectra shown in Figure 5b–d depict various types of crosstalk components. Ideally, these spectra should exhibit minimal transmittance, making them unnecessary. The relatively higher residual polarization crosstalk observed in the PSR is due to insufficient mode conversion in the rib waveguide area and a mode split ratio in the L-ADC area, particularly for the TM_00_-to-TM_00_-mode crosstalk. To address these issues, TM_00_-mode rejection filters, based on cascade-connected SDCs, were serially connected to the output of the PSR.

In this study, we fabricated both single PSR and TM_00_-mode rejection-filter-integrated PSR devices to experimentally identify the crosstalk filtering efficiency. Initially, we characterized the L-ADC-based PSR with parameters identical to those shown in Figure 2a. The measured transmission spectra for output-1 (a) and output-2 (b) are shown in Figure 6, with each condition described in the graphs represented by different colors. Output-1 corresponds to the TE_00_-mode input to TE_00_-mode output signals, while output-2 corresponds to the TM_00_-mode input to TE_00_-mode output signals. Consequently, the other colors represent crosstalk, which ideally should have minimal transmittance. 

Figure 6 shows that each main signal is transmitted consistently through the PSR with a low excess loss. Crosstalk noises are generally suppressed to less than −20 dB, except in the case of the TM_00_-to-TM_00_-mode response. The relatively high crosstalk in the TM_00_-to-TM_00_-mode response is attributed to the insufficient mode conversion efficiency between the TM_00_- and TE_01_-modes in the PSR. This issue could be addressed by optimizing the mode conversion design and selectively removing the residual TM_00_-mode crosstalk using an external rejection filter.

In addition, we performed tests on the S-ADC-based PSR using the same parameters as those shown in Figure 2b. The measured transmission spectra for output-1 (a) and output-2 (b) are shown in Figure 7. The color–output channel relationship shown in Figure 7 is consistent with the description provided in Figure 6. Figure 7 confirms that the S-ADC-based PSR spatially separates arbitrarily polarized signals into two output channels, similar to the TE_00_-mode, as experimentally verified. The spectral characteristics of each main signal and crosstalk component were similar to those observed in the L-ADC-based PSR. Notably, we observed relatively high crosstalk for the TM_00_-to-TM_00_-mode response, which could be due to the adoption of identical hybrid mode conversion regions in both types of PSRs.

Once again, we minimized the polarization crosstalk related to the TM_00_-mode (Figure 6a and Figure 7a). To achieve this, we fabricated and experimentally characterized PSRs integrated with SDC-type TM_00_-mode rejection filters. The SDC parameters used were the same as those shown in Figure 3. The measured spectra at output-1 for both L-ADC-based PSR (a) and S-ADC-based PSR (b) are shown in Figure 8. The color scheme follows the description provided in Figure 6. Figure 8 shows a significant reduction in the polarization crosstalk noises ranging from −25 to −50 dB, compared with the TM_00_-mode polarization crosstalk (TE_00_-to-TM_00_-mode and TM_00_-to-TM_00_-mode responses) depicted in Figure 6 and Figure 7. Despite this crosstalk reduction, the transmission efficiencies for the main signals remained high for both types of PSRs. As the wavelength shifts to the longer-wavelength side, the TM_00_-mode crosstalk rejection ratio becomes significantly higher (Figure 8). Generally, in the Si-nanowire waveguide, the equivalent refractive index tends to decrease on the longer-wavelength side, resulting in improved coupling efficiency for the TM_00_-mode on that side. However, in this situation, the TM_00_-mode crosstalk rejection efficiency trades off with the insertion loss for the main signals because the mode coupling efficiency becomes higher regardless of the polarization state when the wavelength shifts toward the longer-wavelength side. Although the excess loss of the TE_00_-mode at λ = 1.33 μm is not significant (less than 1.5 dB), intentionally lowering the TM_00_-mode crosstalk rejection efficiency could further reduce the excess loss, given that the crosstalk mentioned above is already sufficiently low.

Finally, we characterized the polarization-diversified optical demultiplexer integrated with TM_00_-mode rejection filters (Figure 4b). The mode splitter in the PSR used the L-ADC-type scheme, considering the fabrication accuracy of the foundry process and the area limit of the possible photomask design (Figure 4). Figure 9a shows that the main signals were demultiplexed regardless of their polarization states. Additionally, the transmission of the cross-coupled polarization components was blocked by less than −20 dB across the entire O-band spectral range, similar to the result shown in Figure 5b. Furthermore, experimental results involving the connection of external filter arrays to the PSR (Figure 9c,d) demonstrated a significant reduction of over 20 dB in TM_00_-mode-related polarization crosstalk compared with the results shown in Figure 5c,d.

The demultiplexing spectra and cross-coupled crosstalk shown in Figure 9a,b appear to be somewhat inferior compared with the measured results shown in Figure 5a,b in terms of excess loss, spectral uniformity, and spectral crosstalk. Note that this discrepancy is not due to the presence of TM_00_-mode rejection filters, but rather due to fabrication imperfections of the devices. These imperfections are primarily related to the rib waveguide shape, which is affected by double lithography processes (which impact hybrid mode conversion efficiency), and the channel waveguide shape, which is affected by dry etching processes (which influence the pattern width accuracy compared to the design and fabrication, as well as the magnitude of statistical random phase deviations at each DMZI, depending on the line edge roughness).

The devices shown in Figure 4a,b showed different spectral characteristics due to being fabricated on different SOI substrates at different times. Figure 9a shows that the excess loss at each filtering grid increased as the wavelength shifted toward the longer-wavelength side. This could be due to the MMI width being narrower than the photomask design due to fabrication imperfections. The reduction in the MMI width caused the best split and coupling conditions to shift to the shorter-wavelength side, at approximately λ = 1.26 μm. Meanwhile, the inferior spectral uniformity and higher PDL were attributed to the larger phase errors at each DMZI. We expect that careful control of the fabrication process can address these issues and improve poor spectral repeatability.

Finally, Table 1 depicts a performance comparison of several kinds of the polarization-handling-based CWDM optical demultiplexers. Compared with the previously reported works [13,15,21], the proposed devices exhibited spectral advantages in terms of device footprint, insertion loss, and polarization crosstalk. As stated earlier, it is noted that the reason why the proposed device scheme integrated with the TM_00_-mode rejection filters showed inferior insertion loss and PDL was fabrication imperfections rather than the integration of the TM_00_-mode rejection filters. 

To satisfy practical telecom and datacom applications of integrated transceivers, optical demultiplexers with a flat-topped filter response are desirable. This facilitates easier operation control for laser sources. Such a response can be achieved by connecting additional DMZIs with various specific coupling ratios and optical path differences to modify the transfer matrices of the DMZI-type filters [10,13,22,32].

## 4. Conclusions

We proposed a Si-nanowire-based polarization-diversified CWDM optical demultiplexer, which was theoretically analyzed and experimentally demonstrated. The demultiplexer consisted of the TM_00_-mode rejection-filter-integrated hybrid mode conversion-type PSR and four-channel DMZI-type optical filters. Several types of devices were fabricated using the 200 mm SOI waferscale foundry process. These devices were then characterized using an optical vector analyzer to determine the spectral response from arbitrary polarization inputs to the corresponding outputs of the demultiplexer.

The operation of the fabricated PSRs was successfully demonstrated for both L-ADC- and S-ADC-based mode splitters, demonstrating low excess loss and polarization crosstalks. The integration of the PSR with two identically designed DMZI-type optical demultiplexers confirmed a CWDM-like demultiplexing filter response, irrespective of the input polarization state. The validation resulted in low insertion loss (less than 1.0 dB), PDL (less than 1.0 dB), and interchannel imbalance (less than 1.0 dB), except for some portion of TM_00_-mode-related polarization crosstalk (more than −15 dB).

The experimentally measured polarization crosstalk, which was relatively higher in the TM_00_-mode, was effectively suppressed by a factor of 100 through the connection of the SDC-type TM_00_-mode rejection filter array. Additionally, the four-channel demultiplexing spectral response was verified by integrating the external filter with two identically designed DMZI-type optical demultiplexers. The TM_00_-mode-related polarization crosstalk from arbitrary polarization inputs was further reduced by more than 10–20 dB in the O-band spectral range, resulting in improved detection efficiency for the optical receiver and a lower power penalty in the CWDM optical links.

## Figures and Tables

**Figure 1 nanomaterials-13-02382-f001:**
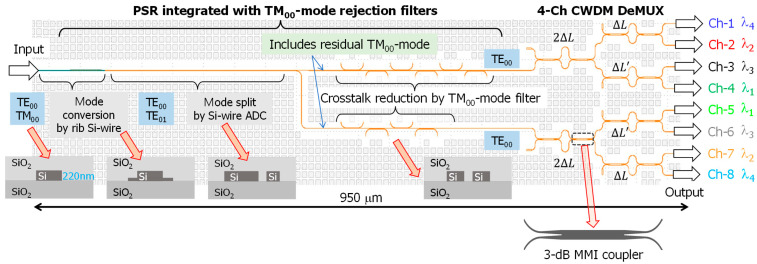
Schematic of Si-nanowire-based polarization-diversified CWDM optical demultiplexer integrated with TM_00_-mode rejection filters.

**Figure 2 nanomaterials-13-02382-f002:**
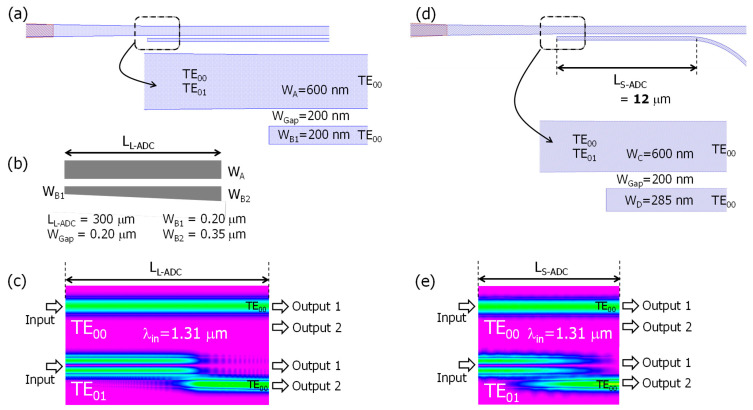
Structures of the mode split regions in the PSR and their propagation characteristics: (**a**) top view of mode split area based on long-length asymmetric directional coupler (L-ADC), (**b**) specification of linear-taper-based L-ADC, (**c**) numerical simulation of optical intensity profile along L-ADC region (λ_in_ = 1.31 μm), (**d**) top view of mode split area based on short-length ADC (S-ADC), and (**e**) numerical simulation of optical intensity profile along S-ADC region (λ_in_ = 1.31 μm).

**Figure 3 nanomaterials-13-02382-f003:**
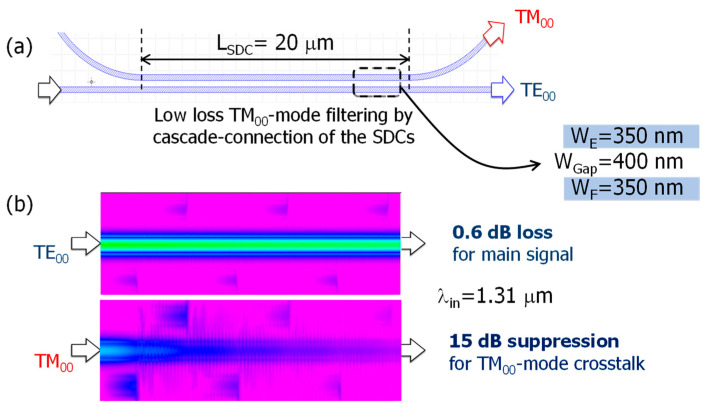
(**a**) Schematic of Si-nanowire-type TM_00_-mode rejection filter based on symmetric directional coupler (SDC), (**b**) numerical simulation of optical intensity profile along sixth cascade-connected SDC region (λ_in_ = 1.31 μm).

**Figure 4 nanomaterials-13-02382-f004:**
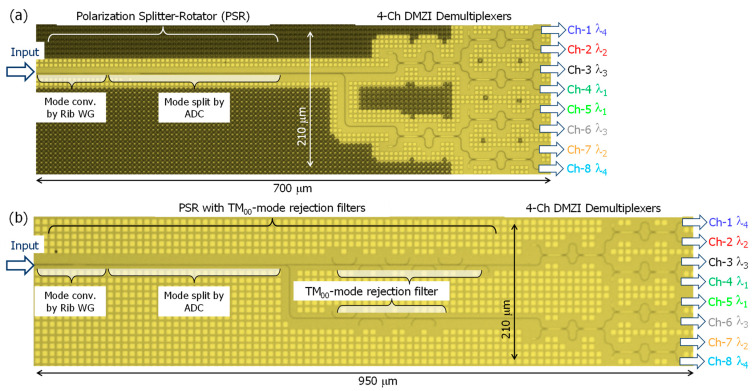
Top view of the fabricated polarization-diversified Si-nanowire CWDM optical demultiplexers without the TM_00_-mode rejection filter (**a**) and with the TM_00_-mode rejection filter (**b**).

**Figure 5 nanomaterials-13-02382-f005:**
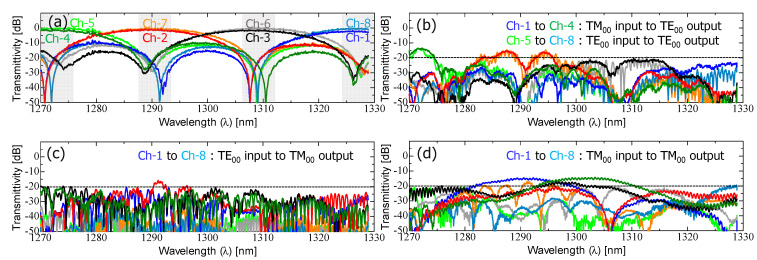
Measured spectral characteristics of the fabricated device shown in Figure 4a: (**a**) main signals (TE_00_-mode input to TE_00_-mode output for Ch-1–Ch-4, TM_00_-mode input to TE_00_-mode output for Ch-5–Ch-8), (**b**) cross-coupled crosstalk (TM_00_-mode input to TE_00_-mode output for Ch-1–Ch-4, TE_00_-mode input to TE_00_-mode output for Ch-5–Ch-8), (**c**) TE_00_-to-TM_00_-mode polarization crosstalk (TE_00_-mode input to TM_00_-mode output for Ch-1–Ch-8), and (**d**) TM_00_-to-TM_00_-mode polarization crosstalk (TM_00_-mode input to TM_00_-mode output for Ch-1–Ch-8).

**Figure 6 nanomaterials-13-02382-f006:**
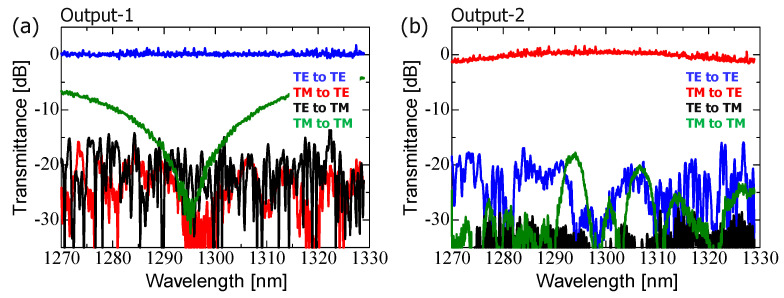
Measured spectral characteristics of the fabricated L-ADC-based PSR shown in Figure 2a at output-1 (**a**) and output-2 (**b**).

**Figure 7 nanomaterials-13-02382-f007:**
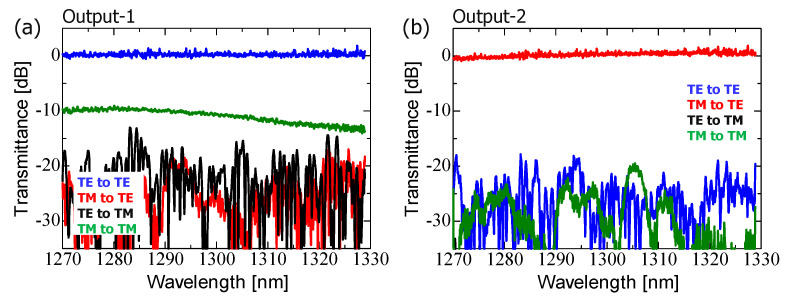
Measured spectral characteristics of the fabricated S-ADC-based PSR shown in Figure 2b at output-1 (**a**) and output-2 (**b**).

**Figure 8 nanomaterials-13-02382-f008:**
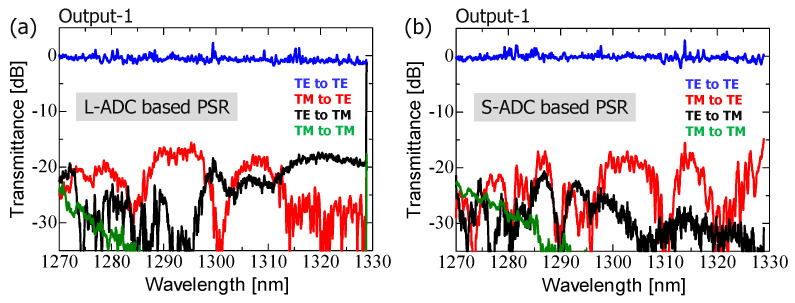
Measured spectral characteristics of the fabricated TM_00_-mode rejection-filter-integrated PSR at output-1: (**a**) L-ADC-based PSR and (**b**) S-ADC-based PSR.

**Figure 9 nanomaterials-13-02382-f009:**
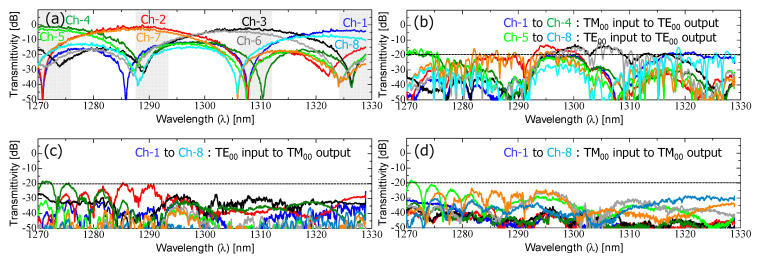
Measured spectral characteristics of the fabricated device shown in Figure 4b: (**a**) main signals (TE_00_-mode input to TE_00_-mode output for Ch-1–Ch-4, TM_00_-mode input to TE_00_-mode output for Ch-5–Ch-8), (**b**) cross-coupled crosstalk (TM_00_-mode input to TE_00_-mode output for Ch-1–Ch-4, TE_00_-mode input to TE_00_-mode output for Ch-5–Ch-8), (**c**) TE_00_-to-TM_00_-mode polarization crosstalk (TE_00_-mode input to TM_00_-mode output for Ch-1–Ch-8), and (**d**) TM_00_-to-TM_00_-mode polarization crosstalk (TM_00_-mode input to TM_00_-mode output for Ch-1–Ch-8).

**Table 1 nanomaterials-13-02382-t001:** Performance comparison for polarization-handling-based CWDM optical demultiplexers.

Device Parameters	This Work ^1^	This Work ^2^	Ref. [15]	Ref. [13]	Ref. [21]
Material	Si-nanowire	Si-nanowire	Si-nanowire	SiN	SiN
Chip size	0.15 mm^2^	0.2 mm^2^	0.3 mm^2^	2.25 mm^2^	1.1 mm^2^
Insertion loss	<1 dB	<2 dB	<2 dB	<3.5 dB	<2 dB
PDL	<1 dB	<3 dB	<0.1 dB	<1.5 dB	<1 dB
λ crosstalk ^3^	<–20 dB	<–15 dB	<–18 dB	<–15 dB	<–25 dB
Pol. Crosstalk ^4^	<–15 dB	<–25 dB	NA ^5^	NA ^5^	NA ^5^
Flat spectra	No	No	Yes	Yes	No
Grid match ^6^	Good	Good	Good	Bad	Bad

^1^ Proposed device scheme without TM_00_-mode rejection filters. ^2^ Proposed device scheme integrated with TM_00_-mode rejection filters. ^3^ Wavelength crosstalk from neighboring channels. ^4^ Polarization crosstalk (TM_00_-mode) from the PSR. ^5^ No experimental data were shown for polarization crosstalk. ^6^ Degree of grid matching from four CWDM channels.

## Data Availability

Data underlying the results presented in this paper are not publicly available at the time of publication, but may be obtained from the authors upon reasonable request.

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
