# Peer review of "Silicon-Nanowire-Type Polarization-Diversified CWDM Demultiplexer for Low Polarization Crosstalk"

_nanomaterials, 2023, doi:10.3390/nano13162382_

Round 1
Reviewer 1 Report
This manuscript presents the design, fabrication, and characterization of a polarization diversified coarse wavelength division multiplexing (CWDM) demultiplexer based on silicon nanowire waveguides. The demultiplexer consists of a polarization splitter-rotator, TM-mode rejection filters, and Mach-Zehnder interferometer filters. The results demonstrate low insertion loss, polarization dependence, and crosstalk, indicating the potential of this approach for optical communications applications. The manuscript is well-written and the results are clearly presented. I have a few suggestions to further improve the manuscript:
1. The introduction could provide more background on the limitations of existing CWDM demultiplexers in terms of polarization dependence. This will help establish the motivation for the proposed design.
2. More information on the design process and rationale behind the various component choices (rib waveguides, ADC, MMI couplers, etc.) would strengthen the manuscript, especially for readers less familiar with silicon photonics device design.
3. The TM-mode rejection filter section is brief – adding more details on the design and optimization would be beneficial.
4. The conclusion could include a more detailed comparison against previous state-of-the-art CWDM demultiplexers and highlight the specific advantages enabled by this design.
5. Carefully proofread the manuscript to check for any grammatical errors or typos.
6. Figures 2 and 3: Consider labeling the waveguides as TE00, TE01 etc. for clarity.
Author Response
Please find the appended file for the reviewer's report.

Reviewer 2 Report
The experimental work has been carefully carried out and the data thus obtained supports the analysis presented.
Author Response
Thank you for reviewing and valueable comments and suggestions.
Reviewer 3 Report
The paper describes both the simulation and experimental results of a silicon nanowire based polarisation diversified CWDM de-multiplier with low cross-talk. My main question is that the authors present the simulations and go into great detail about the dimensions and even mention that there are issues with fabrication variations. However, when they then fabricate these devices, there is no mention at all of the actual dimensions/ distances. All they are referring to are the nominal, ie. expected, values.
A minor comment is that in the paragraph starting at line 154, it appears that the authors are referring to the wrong figure. They are referring to Figure 3, where it probably should be Figure 2. Also the multitude of lines in Figure 5 and Figure 9 make them difficult to read.
Author Response
Thank you for reviewing and valuable comments and suggestions. Please find the appended file for the reviewer's report.
